# Ecohydrological Separation Hypothesis: Review and Prospect

**Yaping Liu [1,2,\*], Yongchen Fang [1], Hongchang Hu [3], Fuqiang Tian [3], Zhiqiang Dong [1] and Mohd Yawar Ali Khan [4]**

[1] College of Resource Environment and Tourism, Capital Normal University, Beijing 100048, China; yongchen.fang@foxmail.com; (Y.F.); dzqcnu@163.com (Z.D.)

[2] Beijing Laboratory of Water Resources Security, Capital Normal University, Beijing 100048, China

[3] Department of Hydraulic Engineering, Tsinghua University, Beijing 100084, China; huhongchang@tsinghua.edu.cn (H.H.); tianfq@tsinghua.edu.cn (F.T.)

[4] Department of Hydrogeology, King Abdulaziz University, Jeddah 21589, Saudi Arabia; yawar.gr44@gmail.com

\* Correspondence: y.liu@cnu.edu.cn

**Abstract:** The ecohydrological-separation (ES) hypothesis is that the water used for plant transpiration and the water used for streams and groundwater recharge comes from distinct subsurface compartmentalized pools. The ES hypothesis was first proposed in a study conducted in the Mediterranean climate region, based on the stable isotope method in 2010. To date, the ES hypothesis has proven to be widespread around the world. The ES hypothesis is a new understanding of the soil water movement process, which is different from the assumption that only one soil reservoir in the traditional hydrology. It is helpful to clear the water sources of plants and establish a new model of the ecohydrological process. However, the theoretical basis and mechanism of the ES hypothesis are still unclear. Therefore, we analyzed the characteristics of ES phenomenon in different climatic regions, summarized the research methods used for the ES hypothesis, concluded the definitions of tightly bound water and mobile water, discussed the mechanism of isotopic differences of different reservoirs and their impacts on ES evaluation and pointed out the existing problems of the ES hypothesis. Future research should focus on the following three aspects: (a) detailed analysis of ES phenomenon characteristics of different plant species in different climatic regions; (b) further understanding of the ES phenomenon mechanism; (c) improvement of the experimental methods.

**Keywords:** ecohydrological separation; stable isotopes; tightly bound water; mobile water; plant water source

## 1. Introduction

Soil water is the primary source of water for crops and natural vegetation and an essential component of the ecohydrological cycle [1,2]. The translatory flow theory assumes that precipitation infiltrates into the soil. Then, the event water (i.e., "new water") immediately mixes with the water initially existing in the soil (i.e., "old water"). Finally, the well-mixed water discharges in the form of evapotranspiration and runoff [3,4]. In other words, the water absorbed and transported by plants, as well as water discharged into groundwater and streams, comes from the same subsurface pool.

However, studies of hydrogen and oxygen stable isotopes of plants and their potential water sources provide new information on the soil and plant water relations [5,6]. For example, the hydrogen isotope ($^2$H) method indicates that the small trees on the riparian utilized stream water as the main source, while mature trees used groundwater as the main water source [7]. Moreover, Brooks et al. [8]

characterized the $\delta^{18}$O and $\delta^2$H values of precipitation, stream water, soil water of different layers and xylem water in a small watershed under the Mediterranean climate. They pointed out that two pools exist in soil, i.e., mobile water pool (MW) and tightly bound water pool (TBW). The two pools are not thoroughly mixed. MW discharged into groundwater or streams, while TBW is used for plant growth. Therefore, the stable isotope values of stream water and plant water are significantly different. This phenomenon is called ecohydrological separation (ES, also known as "two water worlds" hypothesis) [9].

Since then, quite a few studies on the ES hypothesis have been performed in different regions [10–15]. For example, Evaristo et al. [16] analyzed the $\delta^{18}$O and $\delta^2$H data from 47 distributed sites globally and found that the ES phenomenon exists widely in different biomes and global scales (80% sites). However, the ES phenomenon does not exist all year round, and the degree of separation varies with environmental conditions, such as rainfall amount, tree species and soil types. If no mixing occurs between the two soil reservoirs, the ES is completely separated. If partial/complete mixing occurs between the two soil reservoirs, it indicates the existence of ecohydrological connectivity (EC, i.e., incomplete separation) [17,18].For instance, Luo et al. [19] studied the frequency and duration of the ES and pointed out that ES is more likely to occur in spring and winter. Additionally, Nehemy et al. [20] observed the high-resolution water usage pattern of willows. They found that willows used both MW and TBW. The TBW accounted for about 65% of the water used in xylem, indicating that EC exists in the area. Qiu et al. [21] studied stable isotopes of different plant species and their potential water sources in an arid region of western China. It was found that both plant water and soil water were plotted below the local meteoric water line (LMWL) in the $\delta^2$H-$\delta^{18}$O diagram, indicating that there is an ES phenomenon in the study area. Still, the degree of separation varies with different tree species.

The ES hypothesis challenges the theory of translatory flow. Understanding the ES hypothesis and its mechanism helps reveal the specific flow path of water in the basin, further understand the role of soil water in evapotranspiration and runoff process and clarify the process of water absorption by plants and nutrients transport in the soil. The ES hypothesis states that plants are prone to extract fixed, tightly bound water with low water potential in the soil [22,23]. Since MW with high water potential in the soil is quickly discharged and eventually enters the stream, MW is not easily absorbed by plants [24,25]. Plants use less MW for most of the year, not just when MW disappears (e.g., drought period) [16]. It is well known that the transpiration of plants requires much energy to absorb TBW that is tightly bound to the soil matrix [26]. Then it raises a fundamental question about the ES hypothesis. If there is an ES phenomenon exists when more MW are surrounding the roots with stronger mobility, why do plant use the TBW, which is more difficult to obtain? Also, what factors affect ES and what is its mechanism? These are all important issues in ES research. Evaristo et al. [27] argued that ES probably related to the fundamental processes that drive soil drying (such as soil water evaporation, root water uptake and drainage) and have a less seasonal relationship with ecological hydrology. Hervé-Fernández et al. [28] suggested that the ecohydrological connectivity between TBW and MW in the soil may be caused by heavy rainfall events in autumn and winter. Zhang et al. [29] concluded that ES may be related to plant root distribution and soil-depth by studying the water source of apple trees on the loess plateau, China. However, the mechanism of ES is still unclear.

The hydrogen and oxygen stable isotope methods are currently used in ES research. Because MW and TBW have different residence times in soil and are affected by precipitation input and evaporation to varying degrees, it is generally assumed that MW and TBW have different isotope signals. However, McCutcheon et al. [30] found that although MW and TBW have different fluidity, they may have similar isotopic characteristics. In other words, different isotope signals between distinct subsurface pools may be irrelative to the mobility of soil water. If we distinguish TBW from MW by the difference of isotopic signals, it may lead to some errors [30–32]. Thus, it is necessary to understand the assumptions and limitations of common isotope-based analysis methods in ES research.

Therefore, we reviewed the published literature on ES hypothesis with the aim of (a) summarizing the research methods of ES hypotheses and their applicable conditions, (b) revealing the characteristics of ES phenomena in various climatic regions, (c) concluding the definitions of TBW and MW, and (d) discussing the mechanism of isotopic differences in various reservoirs and its influence on ES evaluation. We also summarized the current major issues regarding ES hypothesis research and discussed future research directions.

## 2. Research Methods of ES Hypothesis

A key point of ES hypothesis research is the identification of plant water sources. The method of plant water sources identification based on hydrogen and oxygen stable isotope can be briefly divided into two categories, i.e., traditional method and new method. The traditional method assumes that the plant's water source comes from the same soil reservoir. It usually qualitatively identifies water sources of plants through analyzing the isotope signals of xylem water and its potential water sources, such as precipitation, soil water, groundwater and stream water. Then, models are used to quantitatively calculate the contribution proportion of different sources to plants [33–36]. The widely used models include two/three end-member mixing analysis model, multisource linear mixing model (IsoSource) [37–39], Bayesian mixing model (such as SIAR, MixSIR, MixSIAR) [40–45], etc.

The new method, also known as ES evaluation method, is based on the ES hypothesis, which divides the soil water into two pools: MW and TBW [22]. Similar to the traditional method, the new method also includes qualitative evaluation and quantitative analysis. Qualitative assessment is key to identify whether the ES has occurred in an area. On this basis, model methods (end-member mixing analysis model, etc.) are further used to calculate the mixing degree of two soil reservoirs and the utilization ratio of plants to TBW, etc. Since the models used for quantitative evaluation of ES is same as the traditional method, thus, this paper focuses on the qualitative assessment of ES hypothesis based on hydrogen and oxygen stable isotope methods. The ES evaluation methods mainly include dual-isotope plot diagram ($\delta^2$H-$\delta^{18}$O), line-conditioned excess (lc-excess) and piecewise isotope balance (PIB) method [19].

Dual-isotope plot diagram can be used to determine whether ES phenomenon occurs or not visually. Theoretically, groundwater and stream water are sources from precipitation and they should be plotted on or near the global meteoric water line (GMWL) [46]. However, due to processes such as evaporation and mixing, soil water deviates from the GMWL and varies along the soil water evaporation line (SWEL) [47]. Figure 1 shows the values of $\delta^2$H and $\delta^{18}$O of different water bodies at 11 locations where the ES phenomenon was observed. When the isotopic signals of plant water and soil water are significantly different from stream water and groundwater, that is, they deviate from the GMWL and fall below the GMWL; it can be considered that the ES phenomenon has occurred. Conversely, if isotopes of plant water and soil water show similar signals to groundwater and stream water, that is, they fall on the GMWL, it is considered that no ES phenomenon occurred, or ES phenomenon is not significant.

The dual-isotope plot method is usually applied to assess the ES hypothesis in areas where evaporation effects are significant. It is easy to distinguish the TBW (mainly constituted by evaporated water) from the MW in those areas using the dual-isotope plot method. However, the kinetic non-equilibrium fractionation effect of water isotopes is weak and evaporation has little impact on the composition of water isotopes in humid regions. Therefore, it is difficult to distinguish the isotope signals of different water bodies in the dual-isotope diagrams of humid areas [17,19].

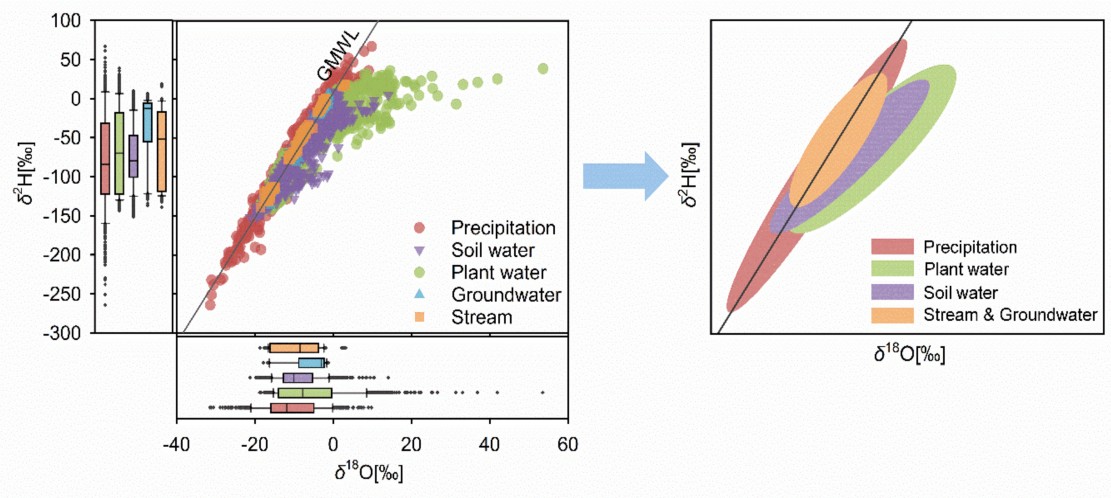

**Figure 1.** Evaluation of ecohydrological-separation (ES) hypothesis based on dual-isotope plot method, i.e., isotopic differences in plant water, soil water, groundwater and stream. Data from [8,11,14,27,30,48–53].

The term of lc-excess is based on the concept of deuterium excess (d-excess) [16,54]. The d-excess quantifies the isotope offset from the GMWL. The lc-excess quantifies the isotopic distance between the water sample and the LMWL. Therefore, the lc-excess is used to quantify the degree of migration of environmental water and precipitation input. The lc-excess values of precipitation, stream water and groundwater usually fluctuate around 0 (Figure 2). In contrast, the lc-excess values of the water bodies undergoing evaporation are negative. The more negative the lc-excess value, the stronger the evaporation of the water sample [17,55]. By comparing the lc-excess value of different water bodies (such as stream water, soil water and plant water), we can distinguish between evaporated water and non-evaporated water [16,21]. The lc-excess is calculated using the following equation:

$$\text{lc} - \text{excess} \ (‰) = \delta^2 H - a \ \delta^{18} O - b \tag{1}$$

where a and b are the slope and intercept of the LMWL.

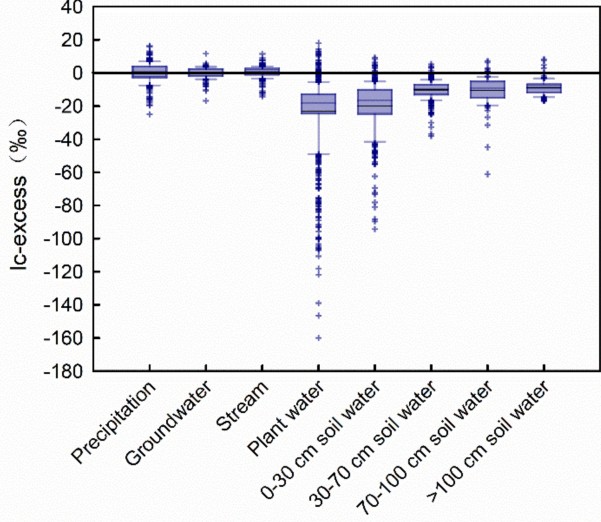

**Figure 2.** Evaluation of the ES hypothesis based on the line-conditioned excess (lc-excess) method, i.e., comparing the lc-excess values of distinct pools. Box plot shows the maximum, upper quartile, average, median, lower quartile and minimum. Data from [8,11,14,27,30,48–53].

As shown in Figure 2, the lc-excess values of groundwater and stream water fluctuate around 0, while the lc-excess values of plant water and soil water are negative. The more negative lc-excess value of plant water and soil water indicates a higher separation degree. In addition, the lc-excess value of soil water was close to zero when the soil-depth increased, indicating that the evaporation of soil water decreased with the increase of soil-depth.

The PIB method uses isotope values of xylem water and precipitation to calculate the contribution ratio of precipitation to xylem water from two consecutive sampling events [19]. If the contribution ratio of precipitation to plant water is small, the PIB method assumes that an ES phenomenon has occurred. Otherwise, if the contrition ratio is large, the PIB method considers that EC has occurred. The advantage of the PIB method is that it can show the occurrence time and the duration of ES phenomenon. In addition, the PIB method has nothing to do with the difference between xylem water (TBW) and groundwater/stream water (MW) stable isotope signals. The disadvantage of the PIB method is that it ignores the effects of evaporation on the potential water source of the plants. Therefore, it is more suitable for humid regions where evaporation has a relatively minor impact on the isotopic composition of environmental water bodies [19,56].

It should be noted that when we conduct ES research, the above research methods may bring different results. Using only one method to determine the degree of ES may result in large uncertainty of the results. Thus, we should evaluate ES from multiple perspectives by combining different methods, such as continuous measurement of precipitation, soil water content and soil/plant water potential.

## 3. Characteristics of ES in Different Climate Regions

We collected the published stable isotope data ($\delta^2$H and $\delta^{18}$O) for different soil layers and LMWL in different regions of the world and calculated the lc-excess value according to Equation (1). Figure 3 shows the variation of lc-excess values of soil water with depth under different climate conditions. We found that in arid or seasonal arid climate regions (such as temperate continental climate), the deviation of the lc-excess value of soil water from 0 is greater than in humid climate regions (such as temperate marine climate). Meanwhile, in arid climates, the depth of the soil layer undergoing evaporation is deep.

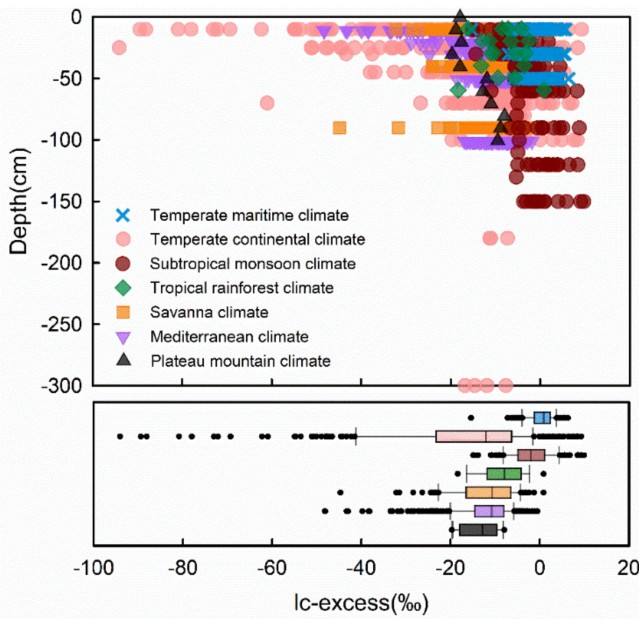

**Figure 3.** Trends of lc-excess values of soil water in different climate types with soil-depth. Data from [5,8,11,13–15,19,21,27,30,49,51,52,57,58].

Table 1 lists the lc-excess values of water bodies in different climate regions. It can be found that in areas with obvious seasonality such as the Mediterranean climate [30–32] and temperate continental climate [33–35], the isotopic values of plant water and soil water are significantly different from those of stream water and groundwater. The lc-excess values of plant water and soil water are more negative than those of stream water and groundwater and the separation degree is higher in these regions. At the same time, the exchange between TBW and MW pools in the soil is more frequent and the EC is better in humid climate regions (such as temperate marine climates) [27,59]. Therefore, the isotope values of plant water and soil water are similar to stream water and groundwater. The lc-excess value is close to 0 and the separation degree in humid climate regions is relatively low [13,57]. In subtropical monsoon climate areas and other areas where it is difficult to distinguish between dry and humid seasons, ES characteristics are more complex. In general, the ES phenomena exist seasonally throughout the world, with higher separation degree in arid season and lower separation degree in the humid season [15,19,57].

**Table 1.** Line-conditioned excess (lc-excess) values of different water bodies in various climate regions.

| Climate | Number of Samples | Water Types | | | | |
|---|---|---|---|---|---|---|
| | | Precipitation | Groundwater | Stream | Xylem | Soil |
| Mediterranean | 537 | 0.2 (3.3) | 1.5 (-) | 0.2 (4.1) | −18.3 (5.9) | −10.7 (6.5) |
| Plateau mountain | 726 | −4.2 (23.9) | 1.7 (3.1) | 2.6 (3.0) | −47.1 (76.9) | −20.1 (33.1) |
| Savanna | 509 | 1.4 (7.9) | 2.7 (0) | −0.3 (3.7) | −23.4 (33.2) | −10.6 (10) |
| Tropical rainforest | 364 | 0.0 (3.2) | 0.5 (4.1) | 2.8 (2.2) | −9.3 (6.7) | −6.3 (5.2) |
| Temperate continental | 766 | 0.9 (7.7) | −0.3 (6.7) | 0.3 (3.1) | −18.6 (11.5) | −12.0 (17) |
| Temperate maritime | 597 | −1.1 (10) | – | 0.4 (1.9) | −8.5 (11.6) | 0.7 (4.5) |
| Subtropical monsoon | 395 | −1.2 (6.5) | −1.1 (3.2) | −5.5 (6.8) | −6.7 (9) | −3.7 (6.8) |

Note: values are median (interquartile range); Data from [8,11,13–15,19,21,27,30,49,51,52,57,58,60].

We also analyzed the relationship between annual precipitation and ES/EC, as shown in Figure 4. It is found that the ES phenomenon exists in both arid regions with annual precipitation 45.7 mm [52] and humid region with annual precipitation 2450 mm [27]. Moreover, ES does not exist all year round. For example, the ES phenomenon was observed only during the period from February to June in the mountainous area of the southwestern United States, the annual precipitation of where is 500 mm [61]. For another example, in the central and southern part of China with an annual rainfall of 1447 mm, the ES phenomenon was observed from March to May and mid-November to mid-January. Still the EC phenomenon was found from mid-January to February [19]. The occurrence periods of ES phenomenon are also different even in regions with similar annual precipitation. For example, in the Vallcebre Research Catchments, NE Spain, with the annual precipitation of about 900 mm, ES phenomenon is occurred from July to September [62,63]. However, EC phenomenon was observed during the period from July to September in Southwest China, with annual precipitation of 826 mm [15]. In general, ES phenomenon were observed under different annual precipitation, indicating the complex impact of precipitation on ES.

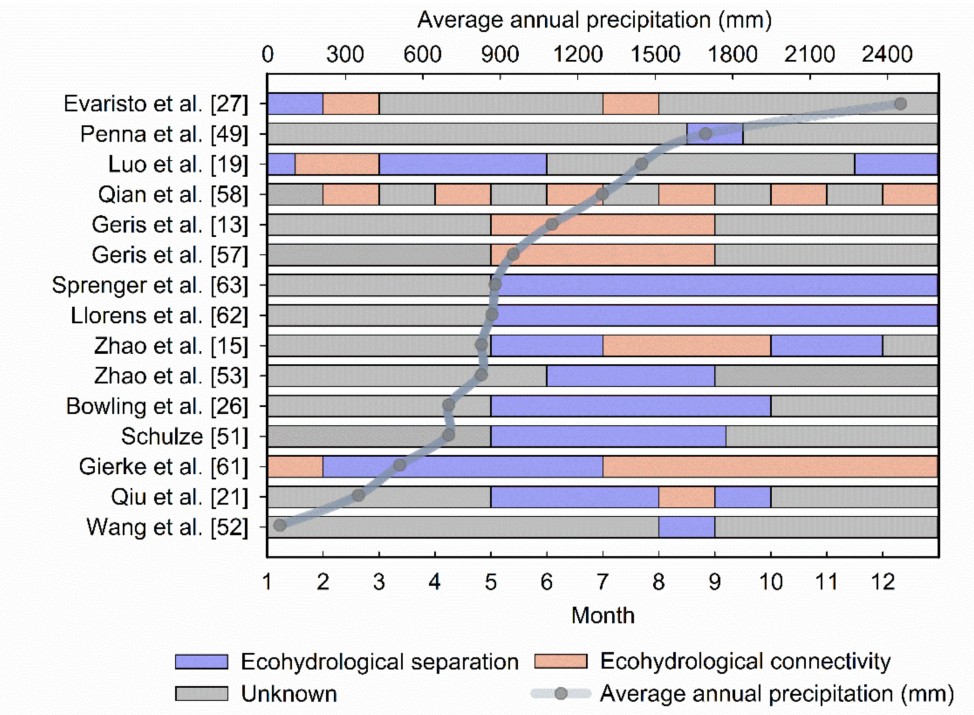

**Figure 4.** Relationship of ES occurrence period and the length of study period of each reference. The average annual precipitation of each study area is also added for reference. Data from [13,15,19,21,26,27,49,51–53,57,58,61–63].

## 4. Definitions of TBW and MW

Figure 5 shows the classification of soil water types based on soil physics and the ES hypothesis. The vertical axis represents soil water suction or soil water matrix potential and the horizontal axis represents soil water content ($\theta$). $\theta 1$–$\theta 4$, respectively represent the corresponding soil water content under different conditions. $\theta 1$ is the permanent wilting point of plants. $\theta 2$ corresponds to the maximum molecular water capacity, that is, the soil water content when the film water reaches the maximum. $\theta 3$ represents the water content at the matrix potential of 65 kPa, which is the average value of the suction range applied by the suction lysimeter. $\theta 4$ is the field water capacity.

Traditional soil physics argues that the bound soil water is composed of hygroscopic water and film water when $\theta < \theta 2$. Hygroscopic water and most of the film water are below the wilting point. It cannot be used by plants and is therefore called non-available water. Although difficult to use, only a small amount of film water above the wilting point can be used by plants. When $\theta > \theta 2$, in addition to hygroscopic water and film water, soil water also includes capillary water, gravity water and groundwater. Plants easily use capillary water held in the soil by capillary action. While gravity water, which drains quickly from the top layer to the bottom layer of the soil after precipitation, cannot be used by plants. Gravity water is also considered as non-available water for plants. The film water (above the wilting point) and capillary water are available water for plants [64–66]. For comparative analysis, the hygroscopic water and film water in the soil physics are regarded as TBW ($\theta < \theta 2$) and the other part is regarded as MW, as shown in Figure 5a.

Based on the ES hypothesis, the classification of soil water types is different from soil physics but is still controversial. It can be generally divided into four categories: (1) TBW is the water that can be retained by the soil and can also be absorbed by plants, which corresponds to the soil available water in soil physics ($\theta 1 < \theta < \theta 4$, Figure 5b) [12,67]. MW corresponds to gravity water; (2) TBW is the soil water that cannot be sampled by the suction lysimeter, but it does not include non-available water in soil physics ($\theta 1 < \theta < \theta 3$, Figure 5c). The mobile soil water content is larger than $\theta 3$ [24]; (3) TBW is the soil water outside the sampling range of the suction lysimeter ($\theta < \theta 3$), including the non-available

water and the rest is MW (θ > θ3) (Figure 5d) [18]; (4) MW corresponds to the gravity water in soil water physics (θ > θ4). All other soil waters (θ < θ4), including hygroscopic water and film water below the wilting point, are considered as TBW (Figure 5e) [17].

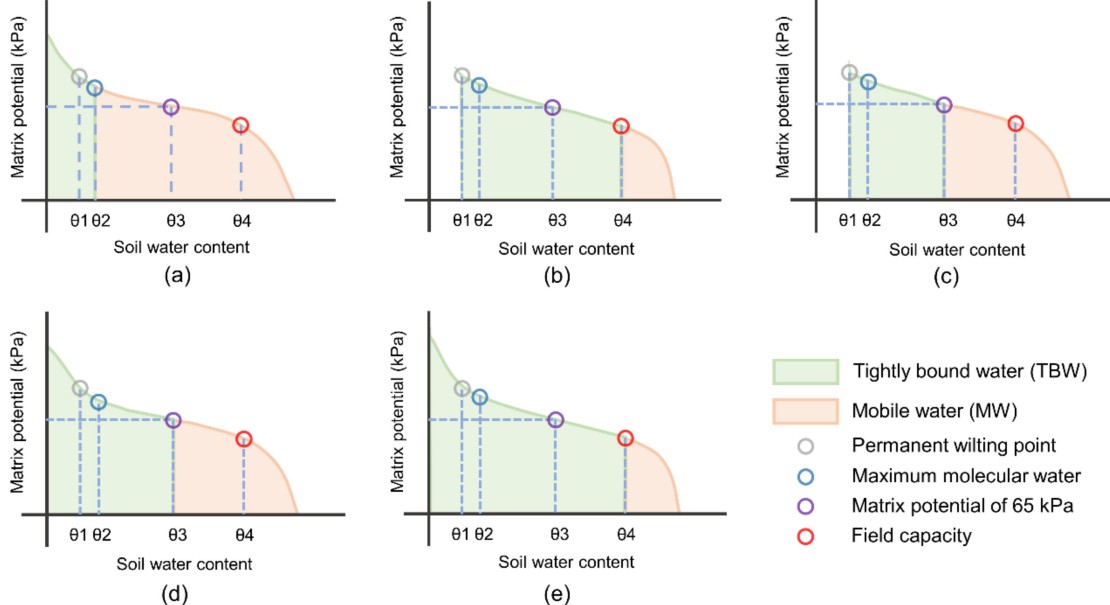

**Figure 5.** Classification of soil water types based on (**a**) soil physics and (**b**–**e**) ES hypothesis. θ1–θ4 represent the corresponding soil water content under different conditions. Figure 5 graphs modified from references. (**a**): [64–66], (**b**): [12,24,67], (**c**): [12,24], (**d**): [18] and (**e**): [17].

## 5. Mechanism of Isotopic Differences in Various Reservoirs and Its Influence on ES Evaluation

The ES hypothesis is proposed and evaluated based on the differences between hydrogen and oxygen stable isotopes in different reservoirs (such as precipitation, plant water, soil water, stream water and groundwater). Understanding the mechanism of isotopic differences in various reservoirs is helpful to understand the formation of ES phenomenon and to evaluate the ES accurately. Plant species, soil types, precipitation events are all factors that impact the isotope signals of different reservoirs. In addition, the experimental methods also affect the observed isotope values, which in turn affect the accuracy of the ES assessment. Therefore, we discussed the mechanism of isotopic differences in different reservoirs and their influence on ES evaluation. The impacts of experimental methods on observed isotopic signals and ES assessment are also discussed.

### 5.1. Plant

Plant water samples' evaporative fractionation signals (lc-excess < 0) are different among different plant species. For example, in semi-arid tropical environments, when water is transferred from xylem to leaves, plant species and their associated leaf phenology (e.g., leaf occurrence time, leaf litter time) are the main factors affecting deuterium enrichment. To prevent water loss, evergreen plants are more prone to reduce water loss and transpiration rate, resulting in less deuterium enrichment from xylem to leaves than deciduous tree species [11]. The isotopic composition and lc-excess value of xylem water of the same plant species in different environments are also different [68]. For example, it was found that the xylem water of *Populus davidiana* growing in different locations have different isotopic characteristics [69].

In addition, various plant species have different root distribution patterns, which can affect the absorption of water by the root system. Recent studies have shown that fine roots play an important role in promoting plant water uptake in arid soils. These fine roots are more likely to come into contact with TBW in soil micropores, resulting in a large contribution of TBW to transpiration [20].

The distribution of root depths could also lead to differences in isotope signals of xylem water of different plants [13]. Shallow-rooted plants mainly absorb shallow soil water and display isotopic signals similar to shallow soil water. Shallow soil water which is affected by evaporations is usually enriched with heavy isotopes of hydrogen and stable oxygen. Thus, water isotopes in the xylem of shallow root plants are also enriched in heavy isotopes. Deep root plants mainly absorb deep soil water, showing similar isotopic signals with deep soil water, i.e., isotopic values are relatively depleted in heavy isotopes. For example, Zhou et al. [70] compared the $\delta^2H$ and $\delta^{18}O$ values of three different desert plants in the same habitat and found that *Reaumuria soongorica* utilizes more surface soil water and its $\delta^{18}O$ value is greater than that of *Nitraria sibirica*. The $\delta^{18}O$ value of *Nitraria sibirica* is greater than that of *Tamarix ramosissima*. Among them, the root system of *Reaumuria soongorica* is shallowest, *Tamarix ramosissima* is deepest and *Nitraria sibirica* is between *Tamarix ramosissima* and *Reaumuria soongorica*. The different isotope signals of xylem water of different plants may be caused by the difference root distribution patterns, rather than the ES phenomenon. Therefore, it is necessary to understand the main reason causing various isotope values of different plants before conducting the ES hypothesis evaluation.

There is no exchange with the external environment when soil water transfer to the plant roots and upwards along the conduit. Therefore, it is generally believed that no isotopic fractionation occurs during water absorption or water transport within trees [71–74]. In other words, the isotopic composition of the plant xylem water can be used to represent the isotopic information of potential water sources of the plant [75–78]. However, there is growing evidence that isotope fractionation occurs at the soil–root interfaces or in plant tissues and isotope fractionation may be more common than previously thought during water uptake in the root system. For example, Geris et al. [13] pointed out that hydrogen isotope fractionation occurs during water uptake and transport in plants. Approximately 3–9‰ hydrogen isotope fractionation was detected at the soil–root interface of halophyte and xerophyte species [79]. Lin and Sternberg [80] found that some coastal halophytes, such as American mangrove plants (*Laguncularia racemosa*, *Rhizophora mangle*, *Avicennia germinans*) and semi-mangrove plants (*Conocarpus erecta*), also experienced significant hydrogen isotope fractionation during the process of water uptake by the root system. Vargas et al. [81] conducted an experiment on *Persea americana*, which shows that isotopic fractionation existed in the process of root water uptake in non-saline and relatively humid environment. Thus, the plot of isotopes of xylem water along the SWEL can also be explained by the isotopic fractionation during the process of root water uptake or water transport in trees. In other words, the isotopic signals of xylem water do not always represent the isotopic signals of plant potential water sources. Therefore, if the isotope fractionation in the process of root water uptake is not considered, it may lead to incorrect identification of plant water sources [68,82,83].

Bark evaporation can affect the isotope composition of xylem water [7,84] and its effect depends on the surrounding environmental conditions and the residence time of the plant xylem water [30]. For example, low temperatures in winter are not conducive to water evaporation. However, plants are usually dormant, resulting in long residence time of stem water in winter. Other factors, such as hydraulic redistribution, are a common physiological phenomenon in plant and also influence the soil water isotope composition [85]. Hydraulic redistribution is likely to be a mix of water between soil and plant, which originated at different points in time and space [24]. For example, under certain circumstances, the root system could release the water absorbed from the deep soil layers into the shallow soil layers (i.e., hydraulic lift). It would result in changes in the isotopic characteristics of the shallow soil water [86], which in turn, will cause changes in the isotopic characteristics of xylem water.

In addition, previous studies have shown that the presence of mycorrhizal fungi can alter the isotopic composition of plant water. The depletion degree of $^2H$ in xylem water of mycorrhizae seedlings is higher than that of non-mycorrhizal seedlings. This is one of the reasons that the isotope value of xylem water in mycorrhizae seedling deviates from LMWL [87].

*5.2. Soil*

Factors such as particle size, porosity and pore size determine the water retention capacity of the soil and then affect the proportion of MW and TBW in the soil. These factors also affect the mixing process of MW and TBW, i.e., how does the pre-event water is replaced by the event water. Theoretically, the finer the soil particles, the better the water retention and the larger proportion of TBW. For example, the particle size of clay is relatively fine. Even after multiple consecutive precipitation events, most (up to 30%) of the pre-event water remains in the pore space. In other words, the proportion of TBW in clay is large [48]. The sandy soil has coarser particles, poor water retention, large porosity, less capillary tension and a relatively small proportion of TBW [88].

The separation degrees of different subsurface pools vary with the soil moisture. Higher soil moisture means that most of the soil pores are occupied by rainwater, and the proportion of MW in the soil is higher. Thus, in the wetter soil, the isotopic differences between MW and TBW is smaller, and the separation degree is lower [13]. In the drier soil, the isotope difference between MW and TBW is greater, and the separation degree is higher [63].

The hydrogen and oxygen isotope values of soil water vary with depth (Figure 6). In general, the isotopic values of soil water decrease with increasing soil-depth, and the spatiotemporal variability of the isotopic composition of soil water decreases. The variation range of $\delta^2H$ and $\delta^{18}O$ values of soil water below 100 cm is lowest. In addition, at different depths, the deviation of the lc-excess of soil water from LMWL is not uniform. The deviation of surface soil is larger than that of deep soil (Figure 2). The lc-excess value of the deep soil is close to 0. Therefore, when conducting ES research, the change of soil water isotope value with soil profile should be considered, and the isotope signal of soil water at different depths should be analyzed. If we only use the isotope value of single-layer soil water or the mixed isotope value of soil water within a certain depth range to compare with the isotope value of stream water and groundwater, it may lead to errors [21].

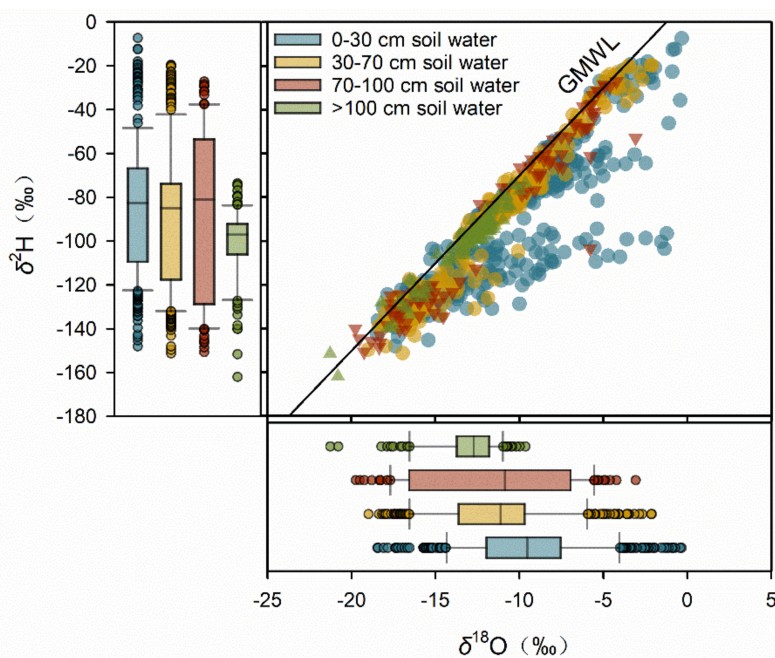

**Figure 6.** $\delta^2H$ and $\delta^{18}O$ signatures for soil water at different depths. Data from [8,14,30,51,52].

Different soil water evaporation conditions will lead to different fractionation processes, which will lead to distinct isotopic signals of MW and TBW. The evaporation of MW occurs under equilibrium condition, i.e., the first-stage evaporation and its isotopic signals are distributed along the LMWL. Whereas, the evaporation of TBW occurs under non-equilibrium condition,

i.e., the second-stage evaporation and its isotopic signals are distributed along the SWEL and away from the LMWL [6,57,89,90].

*5.3. Experimental Method*

Factors such as the study period, sampling frequency and sampling locations, all affect the results of ES analysis. As shown in Figure 4, the ES phenomenon may occur in all months of the year. No direct relationship between the occurrence month of ES phenomenon and the average annual precipitation was found. Currently, most studies on ES focus on the growing season of the plant (May–August). There are few studies on the characteristics of ES throughout the year. High-frequency sampling can help us deepen our understanding of the ES mechanism [22]. If the study period is too short, and the sampling frequency is too low, the details of ES variation with time will be ignored [19]. Thus, it is necessary to increase the duration of the study period and the sampling frequency to understand better the characteristics of ES and its control mechanism. In addition, the choice of sampling location will also affect the analysis results of ES. Watershed is a complex system, in which physical processes (infiltration, evaporation, etc.) occur frequently and vary with location. Even in a very small area, soil moisture is still highly spatially heterogeneous. Therefore, the spatial variability of the isotope values of xylem water and its potential water sources at different sampling locations should also be considered [24,91].

For the same soil or xylem sample, different extraction methods may result in slightly distinct isotopic values of the extracted water. Soil water is usually extracted in situ by suction lysimeter or in the laboratory after collecting soil samples. Laboratory methods for extracting soil water include cryogenic vacuum distillation (CVD), azeotropic distillation, direct vapor equilibration (DVE), centrifugation and high-pressure mechanical squeezing (HPMS), etc. [92–94]. MW mainly composes the extracted soil water by suction lysimeter and the soil water extracted by these laboratory methods consists of TWB and MW (usually a large proportion of TBW). Therefore, the soil water extracted by the laboratory method is usually used to represent TBW, and soil water extracted by suction lysimeter is used to represent MW.

Among these six methods, the CVD method is the most commonly used method for extracting water from the soil and plant xylem [66]. Solid and liquid water can easily vaporize under vacuum. Based on this principle, in the CVD method, the soil/plant sample is heated at high temperature in one end, while the water is collected by condensation with liquid nitrogen in the other end. The advantage of the CVD method is that it is relatively simple to operate and does not require the addition of solvents [95]. Sufficient water can be collected from small samples (<10 g) for stable isotope analysis [96].

The isotope values of soil water extracted by different laboratory methods are also slightly different. For example, the isotopic composition of soil water extracted by centrifugation method and microwave extraction methods is more enriched in heavy isotopes than that extracted by the CVD method and HPMS method [94,97]. Of these four methods, the DVE method may be more accurate for silty soil, but it has the greatest impact on the $\delta^{18}O$ of soil water. No matter what kind of soil, the microwave extraction method has the largest effect on $\delta^2H$ of soil water [94]. Moreover, soil physical properties and soil water content also affect the isotope value of soil water extracted by laboratory methods. For example, if the soil has high clay content and low water content, the isotopic fractionation is more likely to occur when the CVD method is used to extract soil water [98,99]. In addition, it was found that the lc-excess value of soil water extracted by suction lysimeter and CVD method is different and increased with the decrease of soil water content [63].

Plant water is usually obtained by extracting the sampled xylem in the laboratory. The method of extracting plant water is similar to that of soil water, and each extracted method has its applicable range and errors [100]. For example, the HPMS method can quickly extract xylem water, but it can't extract soil water and leaf water quickly [101]. Millar et al. [92] compared six extraction techniques and found that the plant water extracted by centrifugation, microwave extraction and DVE method

was more enriched in $^2$H and $^{18}$O. At the same time, CVD methods (two forms) and the HPMS method result in more depleted values. Millar et al. [92] also pointed out that the DVE method may be the most suitable method for extracting the water of wheat. Therefore, it is necessary to select an appropriate method for different plant and soil types and distinguish whether the isotopic difference between soil water and plant water is caused by the ES phenomenon or different extraction methods.

*5.4. Other Factors*

Precipitation affects soil water conditions, which in turn affects the degree of mixing of MW and TBW in the soil. When there is a long-term drought or less precipitation, the proportion of MW in the soil is lower, and plants tend to use TBW. ES phenomenon is usually obvious [18]. When the rainfall is frequent or long, the soil is usually very wet and even saturated. MW and TBW in the soil become easy to mix, and then the EC increased. At this time, MW and TBW in the soil showed similar isotopic signals, that is, isotopic homogenization of soil water [15,28]. Therefore, it is difficult to observe the ES phenomenon during high precipitation [48,67].

When we conduct ES research, we must thoroughly study all potential sources of water for plants. As we all know, plants absorb soil moisture mainly through their roots. However, there is increasing evidence that plants can also directly absorb dew, fog and atmospheric water through leaves and barks [102–105]. Moreover, rock moisture stored in the weathered bedrock matrix and fractures can also be absorbed by plants, thereby promoting plant transpiration [106,107]. In addition, the flow velocity at the pore-scale of the subsurface water pool will affect the contact time between the water and the mineral surface and then the nutrient concentration in different water pools [63]. The concentration of nutrients in the TBW pool (low fluidity and slow flow rate) is greater than the concentration of nutrients in the MW pool, which may also affect the absorption of TBW by plants [68,108,109].

## 6. Summary and Prospects

At present, ES research is mainly carried out in forest areas [8,14,28,57,63]. ES phenomenon is generally observed in areas with significant seasonal variations in precipitation, such as temperate continental climate [26,30,51,110] and the Mediterranean climate regions [8,49,62,63,111] etc. In humid climates, the performance characteristics of ES is more complex [7,13,57,58], and relatively little work has been done in these regions. Although the ES hypothesis indicates that the water used by plant utilization is mainly derived from TBW in soil, this does not mean that plants do not use MW in soil, as well as stream water and groundwater [27,112]. More work is needed to analyze the ES characteristics and mechanisms for various plants and different soil types in different climatic regions using multiple experimental methods.

How to extract and distinguish TBW and MW from the soil remains an important issue for ES evaluation. Usually, before we can further distinguish between TBW and MW, we must extract water from the soil. However, the process of extracting water from the soil is more complicated than collecting stream water and groundwater [113]. Since there is no method for sampling only TBW, soil water extracted by the CVD method is usually used to represent TBW, and soil water extracted by suction lysimeter is used to represent MW. However, the soil water extracted by the CVD method contains both TBW and MW, so this method is more suitable for soil with low water content [20]. Moreover, the soil water extracted by suction lysimeter method also contains a small part of TBW. To date, some studies mainly collect cryogenic vacuum extracted soil water without collecting MW, which may affect the accuracy of the results. Groundwater and stream water is roughly used to represent MW in soil. However, replacing soil MW with stream water needs to meet the requirement that the main recharge of stream water comes from the runoff formed by precipitation passing through the soil profile. When stream water is mainly recharged by base flow, it cannot approximately represent the soil MW [17]. To better understand the mechanism of the ES, new technologies need to be developed to collect more accurate TBW and MW samples [17,24,114].

The concepts of TBW and MW are still controversial. In addition, some researchers argued that the soil matrix water is better than "immobile", "bound" or "tightly bound" water to describe water that cannot move mobility under gravity, since TBW maybe not fixed in time scale [67,109]. Moreover, the different water-use strategies of trees can change the state of water in the soil, which may be related to species-level control rather than the "boundedness" of soil moisture [67,109].

The ES hypothesis is inconsistent with the theory of plant biology. The ES hypothesis states that even with higher MW, plants will preferentially absorb TBW. However, research has found that plant water consumption responds quickly to precipitation events and shows a significant increase during the rainy season [24]. It is a well-known fact that after rainfall in the rainy season, the water content in the soil increases significantly and contains more MW. Why do plants preferentially use TBW to consume more energy, especially when soil moisture content exceeds the field capacity? In addition, quantitative analysis of the degree of exchange between the two subsurface pools will help us to explore the internal mechanism of the ES.

The ES hypothesis believes that the isotopic difference between plant water and stream water is mainly due to the existence of two distinct subsurface pools. However, the heterogeneity of flow may lead to different flow velocity and path length [115]. That is, the age of water used for plant transpiration in the soil is older than the age of water that supplements stream water and groundwater. Especially in forest areas with high biodiversity, the heterogeneity of water flow is more significant [115]. Using the established hydrological theoretical framework of mass conservation (i.e., different outflow/storage and residence/transit time) and heterogeneity of flow (i.e., different flow path length and flow velocity), the isotopic differences among stream water, groundwater, soil water and plant water also can be explained [115,116]. Therefore, the mechanism of the ES phenomenon is also the focus of future research.

Stable isotopes have been widely used in ecohydrological research, but there are still some uncertainties in the application process [117]. For example, high-precision water source identification for non-wood species (such as grass) still has some limitations [1]. The water vapor cycle under the canopy, that is, water vapor from soil evaporation and tree transpiration, returns to the soil through recondensation, which may cause the distribution of soil water isotopes along the LMWL [110]. In addition, when the water table is shallow, the groundwater with depleted isotopic values is one of the water sources of the plants. The isotopes of the plant are also depleted in heavy isotopes. All these phenomena affect the ES evaluation based on hydrogen and oxygen stable isotope method. However, in general, stable isotope technology providing a practical method for identifying water sources of plants. It is still effective to modify our understanding of soil water movement process. It is necessary to use a stable isotope method combining with other methods in the future, such as high-frequency observation of precipitation, soil water content, plant sap flow, soil/plant water potential to improve the accuracy of ES evaluation.

**Author Contributions:** Y.L. and H.H. designed the study. Y.L., Y.F. and H.H. performed the study and outlined the manuscript. Y.F. collected data and drew figures. F.T., Z.D. and M.Y.A.K. revised the manuscript. All authors have read and agreed to the published version of the manuscript.

**Funding:** This research was funded by the National Natural Science Foundation of China (51879136, 51809173). This research was jointly supported by the National Key R&D Program of China (2017YFC0406006) and the Second Tibetan Plateau Scientific Expedition and Research Program (STEP) (2019QZKK0207).

**Acknowledgments:** We would like to thank Yi Nan from Tsinghua University for his comments regarding this work. We also thank two anonymous reviewers and the Editor for reviewing the manuscript.

**Conflicts of Interest:** The authors declare no conflicts of interest.

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
