# Peer review of "Ecohydrological Separation Hypothesis: Review and Prospect"

_water, doi:10.3390/w12082077_

Round 1

Reviewer 1 Report

Water

Manuscript Number: 835773

Title: Ecohydrological Separation Hypothesis: Review and Prospect

Article Type: Research Paper

Keywords: ecohydrological separation; stable isotopes; subsurface pools; tightly bound water; mobile water

The manuscript Water-835773 focuses on the qualitative evaluation of ES hypothesis based on hydrogen and oxygen stable isotope methods

The work represents a valuable contribution to the literature, but it is lacking in some points. I recommend it for publication after a major revision.

The main trouble points are:

I suggest improving the information and references about stable isotopes behaviour (e.g. Craig H., 1961, Longinelli A. and Selmo E., 2003, Vespasiano et al., 2015)

Chemical composition of waters whit possible relationships between different chemical elements and isotopic composition, whit at least triangular plot and TIS, see for example:

Critelli, T., Vespasiano, G., Apollaro, C., Muto, F., Marini, L., & De Rosa, R. (2015). Hydrogeochemical study of an ophiolitic aquifer: a case study of Lago (Southern Italy, Calabria). Environmental earth sciences, 74(1), 533-543.

Apollaro, C., Fuoco, I., Brozzo, G., & De Rosa, R. (2019). Release and fate of Cr (VI) in the ophiolitic aquifers of Italy: the role of Fe (III) as a potential oxidant of Cr (III) supported by reaction path modelling. Science of the Total Environment, 660, 1459-1471.

It could be useful to improve discussion with further diagrams such as Cl vs d18O and Cl vs. dD. Mobile constituents help comprehension about relationships and evolutions of systems. In this case, to improve knowledge about mixing between different sources.

Add references

Craig, H., 1961. Isotopic variations in meteoric waters. Science 133, 1702e1703.

Longinelli, A., Selmo, E., 2003. Isotopic composition of precipitation in Italy: a first overall map. J. Hydrol. 270, 75e88.

Vespasiano, G., Apollaro, C., De Rosa, R., Muto, F., Larosa, S., Fiebig, J., Mulch, A., Marini, L. (2015). The Small Spring Method (SSM) for the definition of stable isotope - elevation relationships in Northern Calabria (Southern Italy). Applied Geochemistry, 63, 333-346.

Critelli, T., Vespasiano, G., Apollaro, C., Muto, F., Marini, L., & De Rosa, R. (2015). Hydrogeochemical study of an ophiolitic aquifer: a case study of Lago (Southern Italy, Calabria). Environmental earth sciences, 74(1), 533-543.

Apollaro, C., Fuoco, I., Brozzo, G., & De Rosa, R. (2019). Release and fate of Cr (VI) in the ophiolitic aquifers of Italy: the role of Fe (III) as a potential oxidant of Cr (III) supported by reaction path modelling. Science of the Total Environment, 660, 1459-1471.

Author Response

Point 1: I suggest improving the information and references about stable isotopes behaviour (e.g. Craig H., 1961, Longinelli A. and Selmo E., 2003, Vespasiano et al., 2015).

Response 1:

Thank you for your suggestion. We add some information and references in the revised version. See Lines 39-40, Line 121-124. 

The revised sentence is as follows:

Line 39-40. However, studies of hydrogen and oxygen stable isotopes of plants and their potential water sources provide new information on the soil and plant water relations [5,6].

Line 121-124. Theoretically, groundwater and stream water are sources from precipitation, and they should be plotted on or near the global meteoric water line (GMWL) [46]. However, due to processes such as evaporation and mixing, soil water deviates from the GMWL and varies along the soil water evaporation line (SWEL) [47].

Added References

5.Ehleringer, J.R.; Dawson, T.E. Water uptake by plants: perspectives from stable isotope composition. Plant, Cell & Environment 1992, 15, 1073-1082, doi:10.1111/j.1365-3040.1992.tb01657.x.

  1. Sprenger, M.; Leistert, H.; Gimbel, K.; Weiler, M. Illuminating hydrological processes at the soil-vegetation-atmosphere interface with water stable isotopes. Reviews of Geophysics 2016, 54, 674-704, doi:10.1002/2015rg000515.
  2. Craig, H. Isotopic Variation in Meteoric Waters. Science (New York, N.Y.) 1961, 133, 1702-1703, doi:10.1126/science.133.3465.1702.
  3. Clark, I.D.; Fritz, P. Environmental Isotopes in Hydrogeology. New York: Lewis Publishers: 1997; pp. 70-71.

We agree that understanding the stable isotopes behaviours is important for studying the hydrological process. Since this manuscript mainly focuses on the ecohydrological separation evaluation, the detailed stable isotopes behaviours are not introduced in the manuscript. However, we believe that readers could find more information from our references.

Point 2: Chemical composition of waters whit possible relationships between different chemical elements and isotopic composition, whit at least triangular plot and TIS, see for example. It could be useful to improve discussion with further diagrams such as Cl vs d18O and Cl vs. dD. Mobile constituents help comprehension about relationships and evolutions of systems. In this case, to improve knowledge about mixing between different sources.

Response 2: Thank you for your suggestion.

Same as water stable isotopes, chemical elements are good environmental tracers for understanding the hydrological processes. Theoretically, chemical compositions of water also can be used to trace water sources of plant. However, it is difficult to realize due to the limitation of observation techniques.  Besides, the transfer mechanism of chemical elements is more complex than that of water within plans. Currently, stable isotopes play a unique "fingerprint" role in determining the hydraulic connection between different water bodies, and are widely used in the study of plant water source identification. Therefore, hydrochemistry methods were not considered in this study.

Reviewer 2 Report

The manuscript "Ecohydrological Separation Hypothesis: Review and Prospect", by Y. Liu, Y. Fang, H. Hu, F. Tian, Z. Dong, M. Yawar and A. Khan evaluates the use of the stable isotopes of oxygen and hydrogen as tools to check the ecohydrological separation hypothesis. Following an introduction presenting the hypothesis, and the required definitions, the authors critically evaluate the applicability of the hypothesis from the viewpoints of the identification of the plants water sources and their isotopic differences, the influence of climatic variations and the mechanisms that allow and hinder associating isotopic ratios to specific water reservoirs. Special attention is placed onto the influencie of alternative water extraction methods on the isotopic values obtained, and how these may modify the conclusions derived from them.

Overall the subject addressed is of general interest to the scientific community, and while not original work, the manuscript provides a good starting point for anybody considering using oxygen and hydrogen isotopes of plant and soil to gain ecological information, facilitating plenty of references to original work that may be of use. Personally, I found the discussion on the effect of different extraction methods on isotope values most interesting.

As such, in my opinion, the manuscript deserves to be published.

However, before that can be so, some concerns dealing with the use of the English language need to be addressed. While a general revision by a knowledgeable English speaker is required, and as such I will not go into specific details, there are some instances that I cannot but highligh, since they affect the correct understanding of what the authors really meant:

- It is quite frequent to find phrases that have the same verb twice, although in different forms, or have other type of redundancies.

As an example, take the phrase in L 365-366: "There is no direct relationship between the occurrence month of ES phenomenon and the average annual precipitation has been found." My suggestion would be either "There is no direct relationship between the occurrence month of ES phenomenon and the average annual precipitation", or else "No direct relationship between the occurrence month of ES phenomenon and the average annual precipitation has been found".

These, and many others, need to be corrected.

- L 105-107: ... "The traditional method is to direct clarify the potential water sources of plants based on isotope signals, and then calculate the contribution proportion of different sources to plants using models [31-34]".

This phrase makes no grammatical sense, and I do not understand it, nor can figure out what the authors meant.

- L 111: "... further distinguishes the parts used by soil water (MW/TBW) plants, ...".

Please consider rewording. Confusing!

- L 181-184; Fig. 3: This figure is quite difficult to read: too many similar data points that are difficult to discern from each other.

- L 200-202; Table 1: the table heading and the main table should fit into the same page.

- L 389-391: "According to Second Law of Thermodynamics, heat can be transferred from high to low temperatures, thereby quickly and efficiently to extracting water from plants and soil".

This phrase makes no grammatical sense. Please, rephrase!

- 410-411: "At the same time, CVD methods (two forms) and the HPMS method are more depleted".

This phrase makes no scientific sense. Please, rephrase: "At the same time, CVD methods (two forms) and the HPMS method result in more depleted values".

Overall, after reading the manuscript I felt very unsure regarding the actual applicability of Ecohydrological Separation using stable isotopes, and would have liked the authors clearly taking part for or against, together with the relevan arguments to do so.

Author Response

Point 1: Some concerns dealing with the use of the English language need to be addressed. A general revision by a knowledgeable English speaker is required. It is quite frequent to find phrases that have the same verb twice, although in different forms, or have other type of redundancies.

Response 1:

We appreciate the chance to revise the manuscript, so that quality and readability of the manuscript is further improved. All the suggestions have been well taken, and the authors tried to improve the manuscript both structurally and grammatically to the best of our abilities. The language of the whole text has been improved with the help of other colleagues (few native English speaking). Also, the manuscript has been reorganized for a better understanding of the readers.

Point 2: As an example, take the phrase in L 365-366: "There is no direct relationship between the occurrence month of ES phenomenon and the average annual precipitation has been found." My suggestion would be either "There is no direct relationship between the occurrence month of ES phenomenon and the average annual precipitation", or else "No direct relationship between the occurrence month of ES phenomenon and the average annual precipitation has been found".

Response 2: Thanks, revised.

The revised sentence is as follows:

Line 372-373. No direct relationship between the occurrence month of ES phenomenon and the average annual precipitation has been found.

Point 3: These, and many others, need to be corrected.

- L 105-107: ... "The traditional method is to direct clarify the potential water sources of plants based on isotope signals, and then calculate the contribution proportion of different sources to plants using models [31-34]".

This phrase makes no grammatical sense, and I do not understand it, nor can figure out what the authors meant.

Response 3: Thanks, revised.

The revised sentence is as follows:

Line 104-107. It usually qualitatively identifies water sources of plants through analyzing the isotope signals of xylem water and its potential water sources, such as precipitation, soil water, groundwater and stream water. Then, models are used to quantitatively calculate the contribution proportion of different sources to plants.

Point 4: - L 111: "... further distinguishes the parts used by soil water (MW/TBW) plants, ...".

Please consider rewording. Confusing!

Response 4: Thanks, revised.

The revised sentence is as follows:

Line 110-111. The new method, also known as ES evaluation method, is based on the ES hypothesis, which divides the soil water into two pools: MW and TBW.

Point 5: - L 181-184; Fig. 3: This figure is quite difficult to read: too many similar data points that are difficult to discern from each other.

Response 5: Thanks, Fig. 3 has been revised.

Point 6: - L 200-202; Table 1: the table heading and the main table should fit into the same page.

Response 6: Thanks, revised.

Point 7: - L 389-391: "According to Second Law of Thermodynamics, heat can be transferred from high to low temperatures, thereby quickly and efficiently to extracting water from plants and soil".This phrase makes no grammatical sense. Please, rephrase!

Response 7: To avoid ambiguity, the sentence has been deleted.

Point 8: - 410-411: "At the same time, CVD methods (two forms) and the HPMS method are more depleted".This phrase makes no scientific sense. Please, rephrase: "At the same time, CVD methods (two forms) and the HPMS method result in more depleted values".

Response 8: Thanks, revised.

The revised sentence is as follows:

Line 416-417. At the same time, CVD methods (two forms) and the HPMS method result in more depleted values.

Point 9: Overall, after reading the manuscript I felt very unsure regarding the actual applicability of Ecohydrological Separation using stable isotopes, and would have liked the authors clearly taking part for or against, together with the relevant arguments to do so.

Response 9: Thanks. Good suggestion. We state our view on the applicability of stable isotope method on ecohydrological separation in the end of the manuscript.

The revised sentence is as follows:

Lines 494-499. But in general, stable isotope technology providing a practical method for identifying water sources of plants. It is still effective to modify our understanding of soil water movement process. It is necessary to use a stable isotope method combining with other methods in the future, such as high-frequency observation of precipitation, soil water content, plant sap flow, soil/plant water potential to improve the accuracy of ES evaluation.

Round 2

Reviewer 1 Report

it'ok